# Evolution of STAT2 resistance to flavivirus NS5 occurred multiple times despite genetic constraints

Ethan C. Veit [1], Madihah S. Salim[1], Mariel J. Jung[1], R. Blake Richardson[1], Ian N. Boys[2,3], Meghan Quinlan[2,3], Erika A. Barrall[1], Eva Bednarski[1], Rachael E. Hamilton[1], Caroline Kikawa[4,5,6], Nels C. Elde [2,3], Adolfo García-Sastre [1,7,8,9,10,11] & Matthew J. Evans [1] ✉

Zika and dengue virus nonstructural protein 5 antagonism of STAT2, a critical interferon signaling transcription factor, to suppress the host interferon response is required for viremia and pathogenesis in a vertebrate host. This affects viral species tropism, as mouse STAT2 resistance renders only immunocompromised or humanized STAT2 mice infectable. Here, we explore how STAT2 evolution impacts antagonism. By measuring the susceptibility of 38 diverse STAT2 proteins, we demonstrate that resistance arose numerous times in mammalian evolution. In four species, resistance requires distinct sets of multiple amino acid changes that often individually disrupt STAT2 signaling. This reflects an evolutionary ridge where progressive resistance is balanced by the need to maintain STAT2 function. Furthermore, resistance may come with a fitness cost, as resistance that arose early in lemur evolution was subsequently lost in some lemur lineages. These findings underscore that while it is possible to evolve resistance to antagonism, complex evolutionary trajectories are required to avoid detrimental host fitness consequences.

Antagonism of the type I interferon (IFN) pathway in a vertebrate host is required for flaviviruses to establish a productive infection that can induce disease. All tested vector-borne flaviviruses use the nonstructural protein 5 (NS5) protein, which is also the viral polymerase and methyltransferase, as an IFN signaling antagonist[1–4]. While the mechanisms of NS5 antagonism vary between flaviviruses, Signal Transducer and Activator of Transcription 2 (STAT2) is targeted by several, including Zika virus (ZIKV) and dengue virus (DENV)[1,2]. This IFN antagonism is species-specific; STAT2 from *Homo sapiens* is susceptible to NS5 antagonism, whereas *Mus musculus* STAT2 is resistant to

NS5 antagonism, leading to differing infection outcomes with early viral clearance and no pathogenesis seen in immunocompetent mice[1,5–7]. Indeed, infection that is capable of inducing pathogenesis in mice requires either impairing the IFN pathway, or replacing mouse STAT2 with the human gene[7,8].

Previous studies reported that STAT2 in some mammalian lineages, such as primates and rodents, has evolved under positive selection[9], suggesting that it has engaged in one or more host-pathogen genetic conflicts and that STAT2 evolution has impacted the differential susceptibility seen between some primate and rodent

[1]Department of Microbiology, Icahn School of Medicine at Mount Sinai, New York, NY, USA. [2]Department of Human Genetics, University of Utah, Salt Lake City, UT, USA. [3]Howard Hughes Medical Institute, Chevy Chase, MD, USA. [4]Medical Scientist Training Program, University of Washington, Seattle, WA, USA. [5]Department of Genome Sciences, University of Washington, Seattle, WA, USA. [6]Division of Basic Sciences and Computational Biology Program, Fred Hutch Cancer Center, Seattle, WA, USA. [7]Global Health and Emerging Pathogens Institute, Icahn School of Medicine at Mount Sinai, New York, NY, USA. [8]Department of Medicine, Division of Infectious Diseases, Icahn School of Medicine at Mount Sinai, New York, NY, USA. [9]The Tisch Cancer Institute, Icahn School of Medicine at Mount Sinai, New York, NY, USA. [10]Department of Pathology, Molecular and Cell-Based Medicine, Icahn School of Medicine at Mount Sinai, New York, NY, USA. [11]The Icahn Genomics Institute, Icahn School of Medicine at Mount Sinai, New York, NY, USA. ✉e-mail: matthew.evans@mssm.edu

species. However, functional studies have not been conducted to assess the precise impact of STAT2 sequence change across mammalian evolution on flavivirus NS5 antagonism. Understanding STAT2 determinants that confer resistance to flavivirus NS5 antagonism can provide insight into how resistance phenotypes evolve, as well as the barriers to such evolution. Furthermore, given the increasing range of these mosquito-borne flaviviruses worldwide[10], it is important to understand species tropism determinants that influence viral ecology, among them the ability to antagonize the IFN response.

In this article, we found that resistance to flavivirus NS5 antagonism arose multiple times in mammalian evolution. Of note, this does not necessarily mean that current flaviviruses directly drove STAT2 evolution, but rather that one or more STAT2 antagonists that antagonize with flavivirus-like determinants did. Resistance to ZIKV, DENV, and Spondweni virus (SPOV), which we show for the first time also targets STAT2, has arisen on at least two independent occasions in rodents. Additionally, DENV NS5 could not antagonize numerous non-rodent STAT2 proteins, including those of some lemurs and bats. In all species where resistance was evaluated, multiple amino acid changes emerged in a specific order that maintained STAT2 signaling functionality. These data highlight how the interaction of viral antagonists like the NS5 proteins of flaviviruses influenced STAT2 evolution in mammals, the barriers to evolving NS5 resistance, the impacts of STAT2 antagonism on flavivirus species tropism, and offer a road map for overcoming hurdles to the creation of immunocompetent small animal models of flavivirus infection.

## Results

### Mammalian STAT2 susceptibility to NS5 antagonism differs between flaviviruses

To compare STAT2 flavivirus NS5 susceptibility, we used CRISPR/Cas9 to knockout (KO) STAT2 in HEK-293T cells (Fig. 1a). Transfection of luciferase reporter plasmids to measure interferon-stimulated response element (ISRE) promoter activation following IFN stimulation[3] showed IFN did not induce luciferase expression in these cells, but this was restored by co-transfection of a *Homo sapiens* STAT2 expression plasmid (Fig. 1a). Furthermore, signaling in both naïve and complemented STAT2 KO cells was impaired by ZIKV NS5 co-expression. Unless otherwise specified, all references to ZIKV NS5 are from the African lineage MR766 strain. Using human cells permitted the survey of antagonism differences based solely on STAT2 genetics and not other host factors NS5 may need to recruit, such as an E3 ligase to induce STAT2 degradation.

Since only the STAT2 N-terminal (ND) and coiled-coil domains (CCD) interact with NS5 (Fig. 1b)[11], we generated chimeras of *H. sapiens* STAT2 encoding these domains (Fig. 1c) from 37 other mammalian species (Fig. 1d). In STAT2 KO HEK-293T cells, chimeric STAT2 proteins were similarly expressed and able to mediate IFN signaling, though in some with efficiency that was lower than that of *H. sapiens* STAT2 (Fig. 1e, f–h (gray bars)). Our findings demonstrate that African and Asian lineage ZIKV NS5 antagonized all chimeras except the partially resistant camel and koala STAT2 chimeras, and seven of the nine rodent STAT2 chimeras, including the previously reported resistant *M. musculus* STAT2[1] (Fig. 1f and Supplementary Fig. 1a). We additionally found that SPOV NS5 inhibits IFN signaling by targeting STAT2, as STAT2 chimeras exhibited differential susceptibility to SPOV NS5 inhibition (Fig. 1g). While there were slight differences in the degree of antagonism by SPOV NS5 compared to ZIKV NS5, the patterns of susceptibility across species were very similar.

In contrast, STAT2 proteins from more species were resistant to DENV NS5 antagonism, with consistent results across the four dengue serotypes (Fig. 1h, Supplementary Fig. 1b–d). Again, rodents exhibited a broad resistance to antagonism, with only marmot STAT2 exhibiting susceptibility. STAT2 orthologs from several non-rodent species were resistant to DENV NS5, including those from one primate (*Propithecus*

*coquereli*), two ungulates (*Sus scrofa* and *Camelus dromedarius*), one carnivore (*Canis lupus familiaris*), one bat (*Pipistrellus kuhlii*), and one marsupial (*Phascolarctos cinereus*).

### *Mus musculus* STAT2 resistance to ZIKV NS5 is multivalent

By testing the susceptibility of *H. sapiens* chimeras with successively smaller regions of *M. musculus* sequence (Supplementary Fig. 2a), we narrowed the *M. musculus* resistance determinants to ZIKV NS5 to a twelve amino acid region of the CCD (*H. sapiens* amino acids 198–209) that encodes eight amino acid differences between *H. sapiens* and *M. musculus* (Fig. 2a). A chimeric *H. sapiens* STAT2 bearing all eight *M. musculus* amino acids was resistant to ZIKV NS5 antagonism (Fig. 2b, 'M198-209'). Conversely, the *M. musculus* ND/CCD chimera with the *H. sapiens* amino acids 198–209 was fully sensitive to ZIKV NS5 antagonism (Fig. 2b, 'H198-209'). These determinants were also necessary and sufficient to provide resistance to SPOV and DENV NS5 proteins (Supplementary Fig. 2b). Evolutionary analyses using Fast, Unconstrained Bayesian AppRoximation for Inferring Selection (FUBAR)[12] and Phylogenetic Analysis by Maximum Likelihood (PAML)[13] did not find evidence of positive selection at any of the resistance determinant residues, highlighting the importance of our experimental studies with chimeric proteins to dissect the biology of this host-pathogen conflict. Indeed, only one residue (R286 in *M. musculus*) appeared rapidly evolving within rodents, perhaps suggestive of additional host-pathogen conflicts beyond NS5-like antagonists of STAT2 (Fig. 2a, Supplementary Table 1, Supplementary Fig. 3, Supplementary Data 1).

Resistance to antagonism correlated with the loss of ZIKV NS5 binding. As previously reported[1], *H. sapiens* STAT2, but not the *M. musculus* ND/CCD chimera, co-immunoprecipitated with a FLAG epitope-tagged ZIKV NS5 (Fig. 2c). Using this same assay, we found that ZIKV NS5 did not interact with the resistant *H. sapiens* STAT2 mutant with the mouse sequence at amino acids 198–209 (Fig. 2c, '*H. sapiens* + M198-209'), whereas it did interact with the susceptible *M. musculus* ND/CCD chimera with *H. sapiens* residues 198–209 (Fig. 2c, '*M. musculus* + H198-209'). Individual amino acid substitutions between *H. sapiens* STAT2 and the *M. musculus* STAT2 ND/CCD chimera did not further refine our understanding of these determinants, as none exhibited a susceptibility switch (Fig. 2b). However, the susceptibility of many of these mutants could not be assessed because they failed to mediate IFN signaling, which coincided with a loss of protein expression or stability (Fig. 2b). These intragenic incompatibilities suggest that there are substantial fitness barriers for *H. sapiens* STAT2 to evolve resistance in the same manner as *M. musculus* STAT2. For human STAT2 to acquire the same means of ZIKV NS5 resistance as *M. musculus* STAT2, amino acid substitutions would need to be acquired in a specific order that avoids intermediate STAT2 variants that cannot signal.

It is also striking that multiple amino acid changes were required for resistance. We previously showed that single STAT2 amino acid changes (F175A or R176) can abrogate ZIKV NS5 binding[11]. However, these changes also disrupt the ability of STAT2 to mediate IFN stimulated gene expression by affecting the interaction of STAT2 with Interferon Regulatory Factor 9 (IRF9)[11], which mostly binds the top portion of a helix-loop-helix motif of the STAT2 CCD (Fig. 2a, d). In contrast, the *M. musculus* STAT2-ZIKV NS5 resistance determinants (residues 198–209) are located on the lower helix contacting the NS5 methyltransferase (Fig. 2d), and therefore may not impact IRF9 binding. Thus, *M. musculus* resistance to flavivirus NS5 antagonism involves multiple amino acid changes at the NS5-STAT2 interface that avoid critical IRF9 contact sites.

### NS5 antagonism resistance determinants differ among rodent species

We found that rodent STAT2 protein sequences varied at the eight residues that were necessary and sufficient to provide *M. musculus*

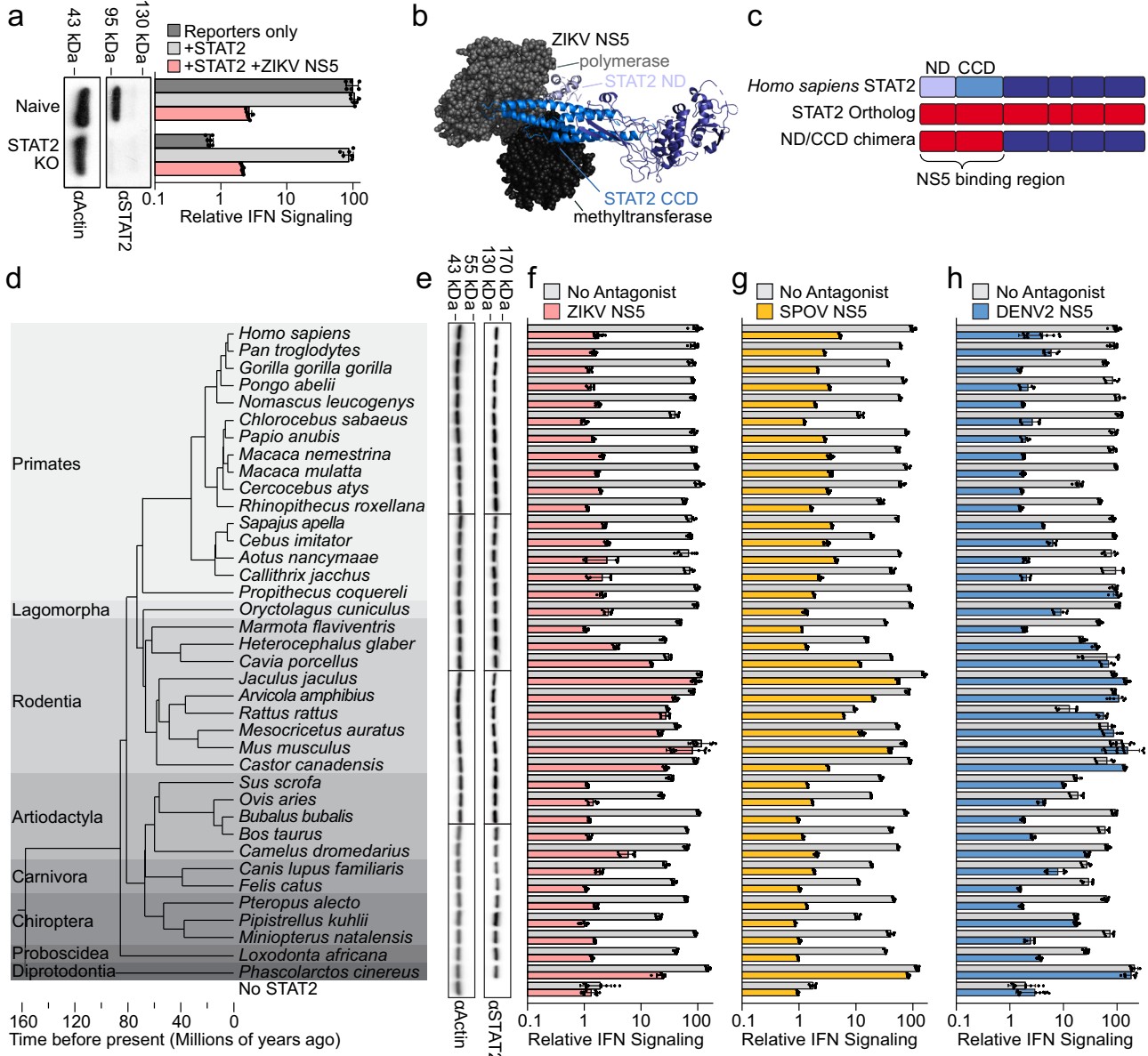

**Fig. 1 | STAT2 susceptibility to flavivirus NS5 antagonism varies among mammals. a** Relative IFN signaling (mean ± s.d., n = 6) (defined as the ratio of activities of the two luciferases normalized to naïve HEK-293's transfected with reporters only) in naïve and STAT2 KO (confirmed via immunoblot with molecular weight marker positions labeled) HEK-293T cells transfected with reporter plasmids only (dark grey), plus STAT2 plasmid (light grey), or plus STAT2 and ZIKV NS5 plasmids (pink). **b** Structure of the complex between ZIKV NS5 (grayscale space filling representation) and *H. sapiens* STAT2 (ribbon representation with domain colored as in (**c**)) (composite of PDB:6WCZ, 6UX2). **c** Schematic of the strategy for the cloning the ND and CCD of various STAT2 orthologs (red fill) as chimeras with the rest of *H. sapiens* STAT2. **d** Node dated molecular phylogeny showing species relatedness of 38 mammalian species. Branch length corresponds to divergence time in millions of years with a scale included below. Grey-scale background coloring corresponds to order classification. **e** STAT2 ortholog ND/CCD chimera expression was compared by immunoblot at 48 h post transient transfection in STAT2 KO HEK-293T cells. Molecular weight marker positions are labeled on the top. **f–h** Relative IFN signaling (defined in this and all subsequent experiments as the ratio of activities of the two luciferases normalized to *H. sapiens* STAT2 in the absence of an antagonist) mediated by each of ND/CCD chimeras in the absence of a viral antagonist (grey) and in the presence of **f** ZIKV NS5 (pink) (mean ± s.d., for Homo sapiens, Mus musculus, and No STAT2, n = 12, all others n = 6). **g** SPOV NS5 (yellow) (mean ± s.d., n = 6). **h** DENV2 NS5 (blue) (mean ± s.d., for Homo sapiens, Mus musculus, and No STAT2, n = 12, all others n = 6). Source data are provided as a Source Data file. All IFN signaling values are derived from at least two independent experiments each with three technical replicates.

STAT2 with resistance to NS5 antagonism (Fig. 3a). *M. flaviventris* and *H. glaber* were the only tested rodent STAT2 proteins that were sensitive to ZIKV NS5 antagonism (Fig. 1f). These species, as well as the resistant *C. porcellus*, are the most deeply diverged rodent species from *M. musculus*, and contained only one or two changes at these residues (Fig. 3a). This suggests that these residues may be incapable of providing resistance in the context of some rodent STAT2 orthologs, hinting at greater STAT2 resistance and susceptibility phenotype complexity in the rodent lineage than previously understood. The

species ranging from *C. canadensis* to *M. musculus* on this tree were resistant to ZIKV antagonism and contain at least four of the eight *M. musculus* resistance determinants (Fig. 3a). Given the conservation of many of the *M. musculus* resistance determinants, it is likely that each of these species relies on the same residues for resistance to flavivirus NS5.

We next focused on *C. porcellus* STAT2, as it was resistant to ZIKV NS5 antagonism but had identical amino acid 198–209 sequence to the susceptible *H. glaber* STAT2 (Fig. 3a). Furthermore, swapping the

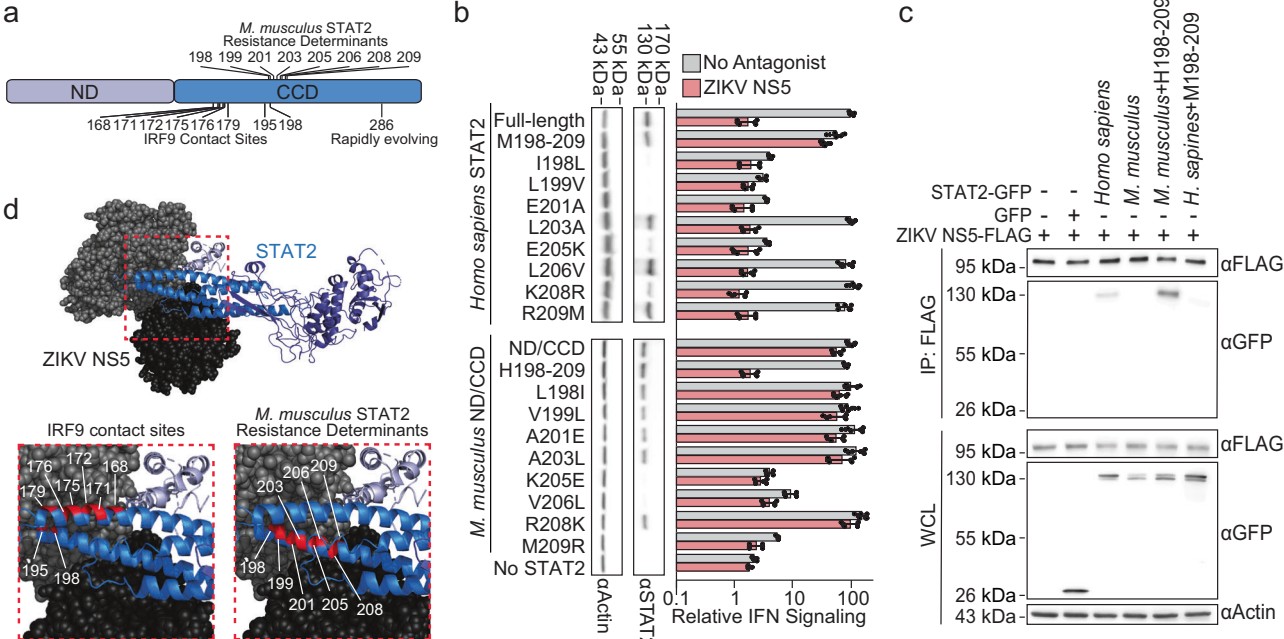

**Fig. 2 | *M. musculus* STAT2 resistance is contained in the coiled-coil domain.** **a** STAT2 ND/CCD schematic showing the locations of the *M. musculus* resistance determinants, IRF9 contact sites, and the residue found to be rapidly evolving in rodents. Numbering is relative to *H. sapiens* STAT2. **b** Relative expression level in absence of antagonist, determined by immunoblot (with molecular weight marker positions on the top) and relative IFN signaling (mean ± s.d., n = 6) mediated by each of indicated STAT2 mutant in the absence of a viral antagonist (grey) and in the presence of ZIKV NS5 (pink). **c** Immunoblots with the antibodies labeled on the right of STAT2 KO HEK-293T cell whole cell extracts (WCE) and FLAG-antibody

immunoprecipitations of the indicated STAT2 proteins with FLAG-tagged ZIKV NS5. To avoid degradation, STAT2 proteins and NS5 were expressed separately, and lysates were mixed prior to immunoprecipitation. Molecular weight marker positions are labeled on the left. Co-immunoprecipitation repeated one other time yielded similar results. **d** Structure of ZIKV NS5 and STAT2, as shown in Fig. 1b, with residues in zoomed images colored in red to indicate IRF9 contact sites and *M. musculus* resistance determinants. Source data are provided as a Source Data file. All IFN signaling values are derived from at least two independent experiments each with three technical replicates.

corresponding residues at these two positions in *C. porcellus* STAT2 with the corresponding *H. sapiens* amino acids (L198I and V199L, *H. sapiens* numbering) resulted in a modest 2-fold increase in ZIKV antagonism (Fig. 3b). While there is a partial overlap with *M. musculus* resistance determinants, *C. porcellus* STAT2 contains additional resistance determinants that are different from all the mouse-like myomorph rodents. These results indicate that resistance to flavivirus NS5-like antagonism was acquired at least twice in rodent evolution.

We mapped the *C. porcellus* STAT2 resistance determinants for ZIKV NS5 antagonism to both the ND (not to single amino acid resolution), and four amino acids in the CCD; V179I, I181A, L198I, and V199L (numbering relative to *H. sapiens* sequence) (Supplementary Fig. 4a–c). Swapping these residues in the *C. porcellus* ND/CCD chimera to the *H. sapiens* sequence rendered it susceptible to NS5 antagonism by ZIKV, SPOV, DENV1, DENV2, and DENV4, but not to DENV3 NS5 (Fig. 3d). While the reciprocal *H. sapiens* STAT2 chimera with the *C. porcellus* resistance determinants (*C. porcellus* ND and I179IV, A181I, I198L, and L199V) was rendered resistant to all four DENV NS5 proteins, it was still susceptible to ZIKV or SPOV NS5, suggesting additional residues in the *C. porcellus* CCD are essential to resistance to these proteins (Fig. 3d). Like *M. musculus* STAT2, *C. porcellus* STAT2 resistance determinants reduced ZIKV NS5 binding, as the full *C. porcellus* ND/CCD STAT2 chimera did not coimmunoprecipitate with ZIKV NS5, but this was restored by adding the *H. sapiens* ND and V179I and I181A mutations (Supplementary Fig. 4d).

Thus, like *M. musculus* STAT2, *C. porcellus* resistance to ZIKV NS5 involves multiple amino acid changes. While the *M. musculus* and *C. porcellus* STAT2 CCD resistance determinants are separated in the linear sequence, they are spatially adjacent to each in the same helix-loop-helix motif of the STAT2 CCD (Fig. 3c). There is overlap in the *C. porcellus* CCD resistance determinants and IRF9 contact sites, but

these resistance determinants avoid the critical residues F175 and R176[11], further highlighting the importance of maintaining IRF9 binding along the pathway to acquiring NS5 resistance.

## Chiropteran STAT2 proteins exhibit monophyletic evolution of resistance to DENV NS5 antagonism

In the above survey, the STAT2 from the bat species *Pipistrellus kuhlii*, but not two other bat species, was resistant to DENV NS5 antagonism. We further explored this observation by testing the STAT2 susceptibility of 12 additional bat species. While all bat STAT2 proteins were susceptible to ZIKV NS5 antagonism, the STAT2 proteins from species in the family *Vespertilionidae* were resistant to antagonism by the NS5 of all four DENV serotypes (Fig. 4a, Supplementary Fig. 5a). This is consistent with a prior study showing robust replication of DENV2 and minimal innate immune response, likely due to STAT2 antagonism, in *Pteropus alecto* cells[14].

By testing chimeras of the resistant *Eptesicus fuscus* STAT2 and the closely related, but susceptible, *Miniopterus natalensis* STAT2, we mapped the *E. fuscus* resistance determinants to eight amino acid differences within residues 283-312 (Fig. 4b, Supplementary Fig. 5b). Adding these eight *E. fuscus* amino acids simultaneously to the *M. natalensis* ND/CCD increases resistance to DENV2 NS5 antagonism, while adding the corresponding *M. natalensis* amino acids to the *E. fuscus* ND/CCD renders it fully susceptible to DENV2 NS5 (Fig. 4b). Swapping these eight changes individually in *M. natalensis* STAT2 to the *E. fuscus* amino acid had no impact on susceptibility to DENV NS5, and four single amino acid swaps greatly impaired in their IFN signaling capacity in the absence of an antagonist, similar to what was seen in single amino acid swaps between *H. sapiens* and *M. musculus* STAT2 (Fig. 4d). Similarly, five of the reciprocal single amino acid changes in *E. fuscus* STAT2 were unable to signal (Fig. 4c). The Q286N and P291E

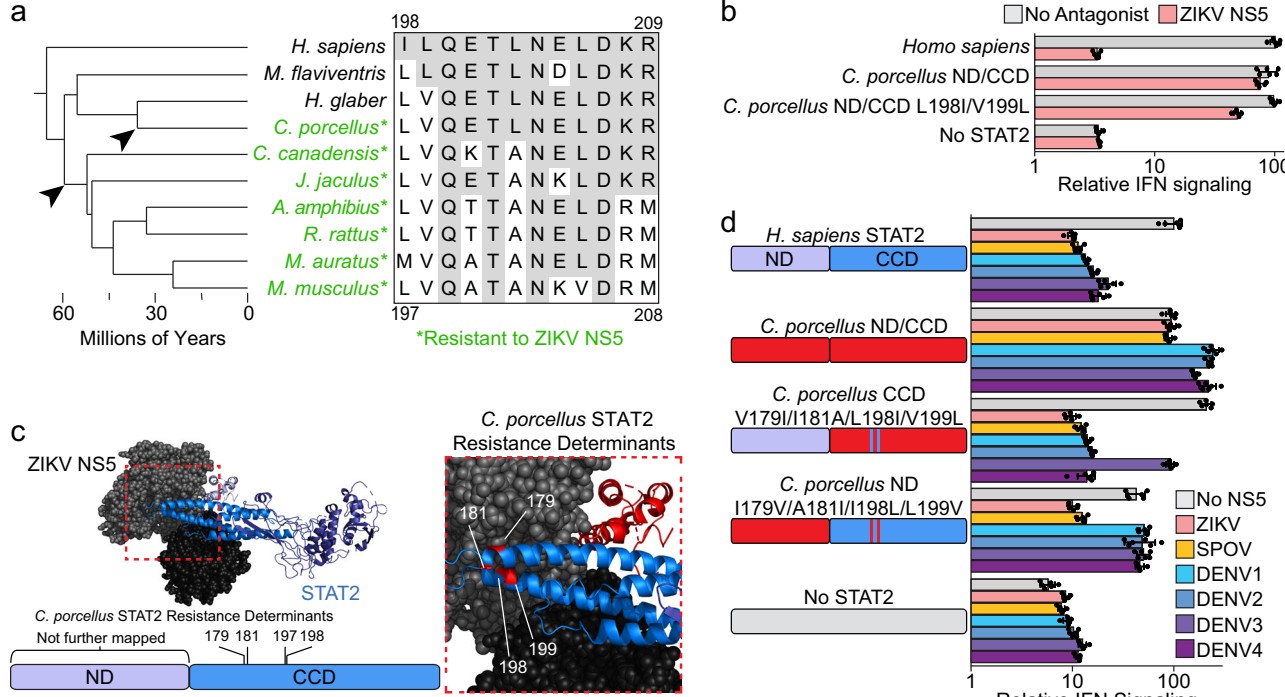

**Fig. 3 | Flavivirus NS5 resistance determinants vary between rodent species.**
**a** Node dated molecular phylogeny on the left showing species relatedness of *H. sapiens* and nine rodents species with those possessing a STAT2 resistant to ZIKV NS5 marked in green (*). Arrows mark the roots of two separate acquisitions of resistance to ZIKV NS5 antagonism. To the right is an amino acid alignment of these species spanning the region encoding the previously mapped *M. musculus* resistance determinants. The numbering on the top and bottom of the alignment corresponds to *H. sapiens* and *M. musculus* STAT2 respectively. **b** Relative IFN signaling (Mean ± s.d, n = 6) of the indicated STAT2 ND/CCD chimeras in the absence (grey) and presence of ZIKV NS5 (pink). **c** Top, space filling representation of ZIKV NS5 in

complex with ribbon representation of *H. sapiens* STAT2 with a zoomed in view of the region encoding the *C. porcellus* resistance determinants (red) to the right. Below is a schematic of the STAT2 ND/CCD showing the *C. porcellus* resistance determinants with numbering corresponding to *H. sapiens* STAT2. **d** Relative IFN signaling (Mean ± s.d, n = 6) of the indicate STAT2 chimeras in the absence of an antagonist (grey), or in the presence of ZIKV MR766 NS5 (pink), SPOV NS5 (yellow), DENV1 NS5 (turquoise), DENV2 NS5 (blue), DENV3 NS5 (light purple), and DENV4 NS5 (purple). Source data are provided as a Source Data file. All IFN signaling values are derived from at least two independent experiments each with three technical replicates.

changes retained activity but had no impact on DENV NS5 resistance (Fig. 4d). However, the T297K change in *E. fuscus* STAT2 modestly reduced resistance to DENV NS5 (2.5-fold), suggesting it is important in the resistance phenotype (Fig. 4d).

FUBAR[12] and PAML[13] analyses of 47 bat species STAT2 ND/CCD sequences found evidence of positive selection at 13 residues (Fig. 4e, Supplementary Table 1, Supplementary Fig. 3, Supplementary Data 2). Some of these residues (Q286, P291, and T297, *E. fuscus* STAT2 numbering) are also resistance determinants. This included the T297 residue, which is the location of the only single amino acid change that had a detectable impact on *E. fuscus* resistance to DENV NS5. Importantly, the *E. fuscus* STAT2 resistance determinants do not overlap with those mapped in rodents, suggesting additional flexibility in the means by which STAT2 can evolve to evade NS5 antagonism. Mapping the *E. fuscus* resistance determinants on the structure of *H. sapiens* STAT2 binding ZIKV NS5, we see most of these residues are located on a flexible loop between two of the helices which make up the four-helix bundle motif of the STAT2 CCD (Fig. 4e). Like rodent STAT2 resistance flavivirus NS5 antagonism, *E. fuscus* resistance requires multiple amino acid changes, with several single changes individually impacting STAT2 functionality.

## Prosimian STAT2 proteins evolved, and then lost resistance to DENV NS5 antagonism

In our above survey, the lemur *Propithecus coquereli* was the only primate whose STAT2 exhibited DENV NS5 resistance. This is striking because non-human primates are essential to the sylvatic cycle of DENV and were previously thought to be pan-sensitive to

infection[15]. We identified four residues, two in the ND and two in the CCD, that were essential for *P. coquereli* resistance to DENV NS5 (Fig. 5a, Supplementary Fig. 6). To further explore lemur STAT2 evolution, we cloned *H. sapiens* STAT2 chimeras encoding the STAT2 ND and CCD of 12 additional lemur species (Fig. 5b). Like *P. coquereli*, all lemur chimeras were susceptible to ZIKV NS5 antagonism (Supplementary Fig. 7a). However, seven were resistant to antagonism by all four serotypes of DENV NS5 (Fig. 5b, Supplementary Fig. 7a). DENV NS5 resistance, which coincides with the appearance of the four determinants mapped in *P. coquereli*, seems to have arisen early in lemur evolution, just after *D. madagascariensis* branches from all other lemurs (Fig. 5c, arrow #1). However, five lemur STAT2 proteins were susceptible despite having all four *P. coquereli* resistance determinants (Fig. 5b, c). The ND and CCD domains of *L. catta* and *P. simus* STAT2 were partially susceptible to DENV NS5, but not to the same degree as *H. sapiens* STAT2 (Fig. 5b, Supplementary Fig. 7a). However, the STAT2 proteins of *E. flavifrons* and *E. rubra* were as susceptible as *H. sapiens* STAT2 to NS5 antagonism by all four DENV serotypes. This suggested that resistance to DENV antagonism was progressively lost in some lemur lineages (Fig. 5c, arrow #2).

We identified two amino acid differences at residues 289 and 290 between the resistant and susceptible lemurs that were responsible for the loss of DENV NS5 resistance (Fig. 5c). Adding both *L. catta* amino acids simultaneously to the resistant *P. coquereli* increased DENV NS5 susceptibility, although not to *H. sapiens* STAT2 levels (Fig. 5d, i). Conversely, swapping these residues in *L. catta* STAT2 rendered it fully resistant (Fig. 5d, ii). STAT2 changes that decreased lemur resistance

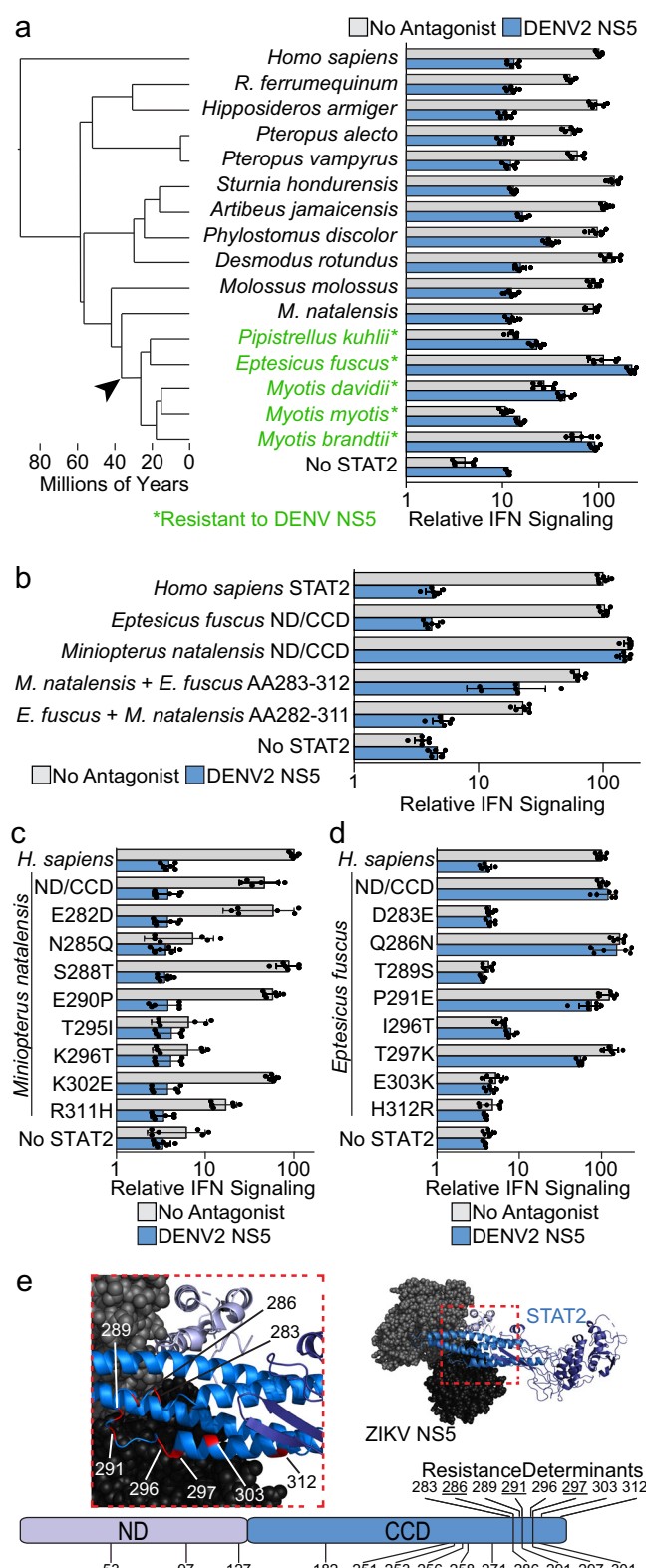

**Fig. 4 | *Vespertilionidae* bats exhibit monophyletic resistance to DENV NS5.**
**a** Node dated molecular phylogeny on the left showing species relatedness of *H. sapiens* and the 15 bat species, with those that have a STAT2 that is resistant to DENV NS5 antagonism marked in green(*). The black arrow indicates the family *Vespertilionidae* branch point where resistance to DENV NS5 antagonism occurred. On the right is a graph corresponding to the Relative IFN signaling (Mean ± s.d, n = 6) of each species ND and CCD chimeras in the absence (gray) and presence of DENV2 NS5 (blue) **b** Relative IFN signaling (Mean ± s.d, n = 6) in the absence (grey) and presence (blue) of DENV2 NS5 for the *M. natalensis* and *E. fuscus* ND/CCD chimeras and chimeras with reciprocal changes in the resistance determinates mapped in (Supplementary Fig. 4b). **c** Relative IFN signaling (Mean ± s.d, n = 6) in the absence (grey) and presence (blue) of DENV2 NS5 for the *M. natalensis* ND/CCD chimera with each of the eight mapped resistance determinants swapped individually to the *E. fuscus* sequence. **d** Relative IFN signaling (Mean ± s.d, n = 6) in the absence (grey) and presence (blue) of DENV2 NS5 for the *E. fuscus* ND/CCD chimera with each of the eight mapped resistance determinants swapped individually to the *M. natalensis* sequence. **e** Top, space filling representation of ZIKV NS5 (as a proxy for DENV NS5) in complex with ribbon representation of *H. sapiens* STAT2. Zoomed region shows *E. fuscus* STAT2 resistance determinants (red). Below, schematic of the STAT2 ND/CCD showing the *E. fuscus* STAT2 resistance determinants, and the residues found to be rapidly evolving using PAML/FUBAR. Underlined numbers indicated resistance determinants that are also rapidly evolving. Source data are provided as a Source Data file. All IFN signaling values are derived from at least two independent experiments each with three technical replicates.

To explore NS5 antagonism in lemur cells, we prepared primary dermal fibroblasts from skin samples obtained from five lemur species. We measured STAT2 degradation following infection with ZIKV and DENV, and as expected, both viruses induced the degradation in *H. sapiens* STAT2 in control Huh-7.5 cells (Fig. 5e, Supplementary Figs. 7b, 8). ZIKV also decreased STAT2 levels in all lemur cells, which is consistent with the ability of ZIKV NS5 to antagonize these species' STAT2 proteins (Fig. 5e, Supplementary Fig. 7b). DENV infection induced the strongest STAT2 degradation in *E. flavifrons* and *E. rufus* cells (Fig. 5d, Supplementary Fig. 7b), which were also the species with STAT2 proteins that were fully susceptible to DENV NS5 antagonism. All three resistant STAT2 lemur species exhibited less STAT2 degradation following DENV infection. While *M. murinus*, and *V. variegata* STAT2 proteins exhibited no degradation, *P. coquereli* levels were moderately reduced.

This suggested that, although less susceptible than other primates, *P. coquereli* STAT2 may not be fully resistant to DENV antagonism. We confirmed this by examining the response of different species STAT2 chimeras to varied quantities of ZIKV or DENV NS5 expression plasmid, which could be used to calculate an inhibitory concentration 50% value (IC50) to denote relative susceptibility to antagonism. *H. sapiens* STAT2 exhibited similar ZIKV and DENV NS5 IC50 values of 12.4 and 15.6 ng per transfection (Fig. 5f, Supplementary Fig. 7c). Conversely, no dose of either NS5 could antagonize *M. musculus* STAT2. ZIKV IC50 values against the *P. coquereli*, *M. murinus*, and *E. flavifrons* STAT2 chimeras were all within 3-fold of *H. sapiens* STAT2, reflecting their generally high susceptibility (Supplementary Fig. 7c). While the *E. flavifrons* chimera was antagonized similarly to *H. sapiens* STAT2, the *M. murinus* chimera signaling was not impacted by any DENV NS5 plasmid dose (Fig. 5f). The *P. coquereli* chimera exhibited intermediate resistance to DENV NS5 with an $IC_{50}$ that was 17.4-fold higher than *E. flavifrons* STAT2. These data paired with the moderate STAT2 degradation following DENV infection indicate that *P. coquereli* STAT2 has an intermediate level of DENV NS5 resistance.

Evolutionary analysis using PAML and FUBAR of the ND/CCD of 12 prosimian species identified several STAT2 residues evolving under positive selection. Expectedly, there was no evidence of positive selection at the four residues we found to be essential to resistance since they were acquired early in lemur evolution and are largely invariant (Fig. 5a, Supplementary Table 1, Supplementary

also had to be acquired in a specific order, as the residue 290 changes on their own rendered STAT2 nonfunctional. The full sensitivity to DENV NS5 antagonism observed in the *Eulemur* genus depended on additional sequence changes at residues 289 and 290. However, swapping the sequences at amino acids 289 and 290 between *L. catta* and *E. flavifrons* was only tolerated with an additional change at residue 127, which on its own did not affect NS5 susceptibility (Fig. 5d, iii, iv).

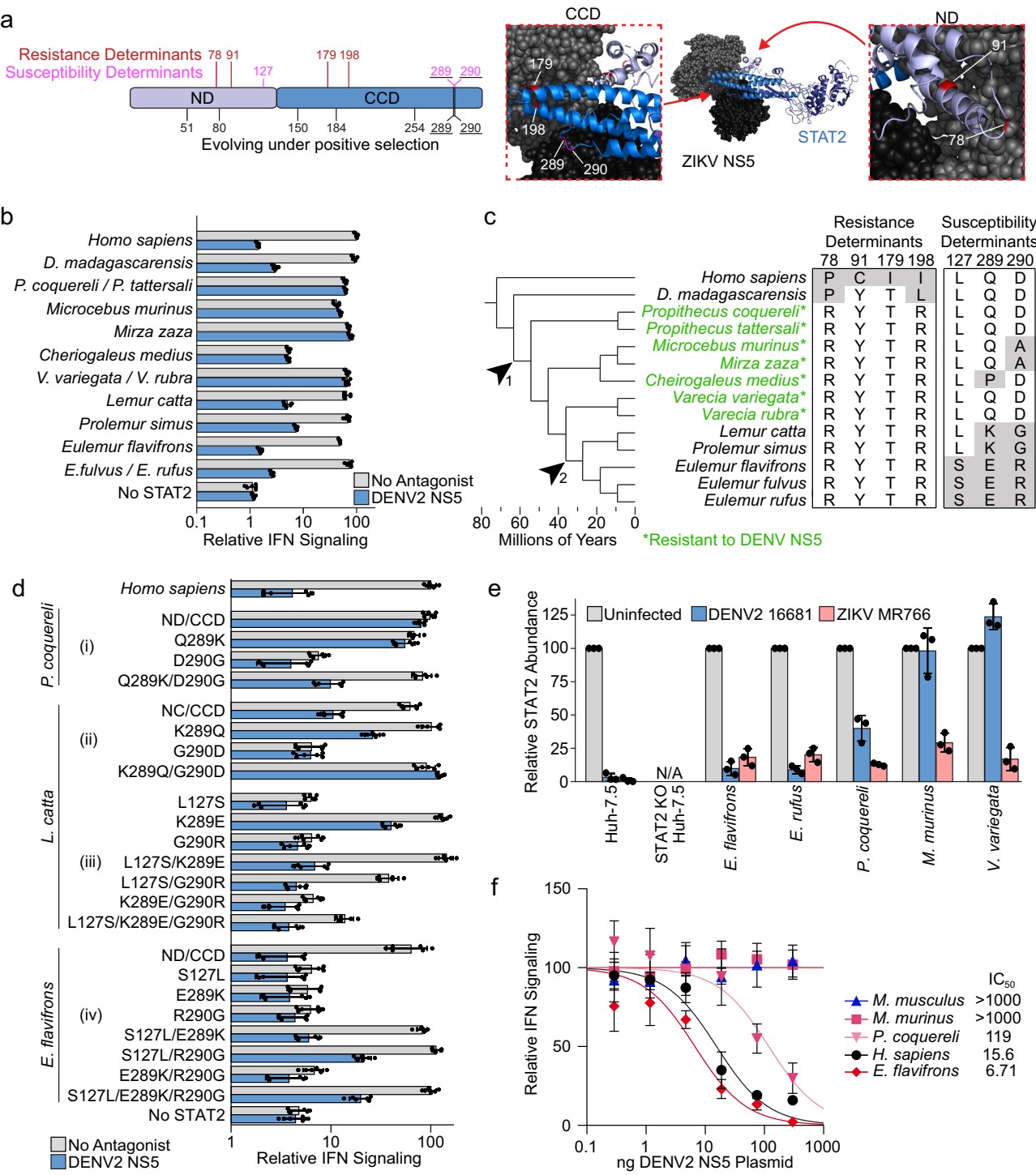

Fig. 3, Supplementary Data 3). However, residues 289 and 290 show evidence of positive selection for both methods. Given the role of these residues in the reversion from resistance to susceptibility, this evidence highlights that there must have been a strong selective force acting on these residues which drove the loss of resistance (Fig. 5a, Supplementary Table 1, Supplementary Fig. 2, Supplementary Data 3).

## Discussion

The ability to antagonize IFN signaling is essential for efficient flavivirus replication and pathogenesis. Flaviviruses are therefore under selective pressure to maintain this activity in species that are relevant for their perpetuation in nature. Conversely, hosts are under

opposing selective pressure to evolve innate immune signaling components that are resistant to such antagonism. Here, we conducted the first survey of STAT2 susceptibility to ZIKV, DENV, and SPOV, which we identified as a STAT2 targeting virus, and found that resistance to flavivirus NS5 antagonism occurred frequently during mammalian STAT2 evolution. While resistance to ZIKV and SPOV NS5 antagonism was largely restricted to rodents, the convergent evolution of resistance utilizing two distinct sets of determinants on this phenotype underscores its importance. DENV NS5 resistance has independently arisen in multiple mammalian orders, suggesting that this viral antagonist is less tolerant to STAT2 sequence variation than the other NS5 proteins surveyed in this study.

**Fig. 5 | DENV NS5 resistance was acquired and subsequently lost in lemur evolution. a** Left, STAT2 ND/CCD schematic with resistance (red), susceptibility (pink), and rapidly evolving residues via PAML/FUBAR (black) marked. Right, space filling representation of ZIKV NS5 (as a proxy for DENV NS5) in complex with ribbon representation of *H. sapiens* STAT2. Zoomed in regions show two views of the DENV NS5 resistance determinants mapped in *P. coquereli* STAT2 (red) and the residues involved in the reversion to susceptibility (pink). Underlined numbers indicated resistance determinants that are also rapidly evolving. Note: STAT2 residue 127 is not resolved in this structure. **b** Relative IFN signaling (Mean ± s.d, n = 6) of STAT2 chimeras in the absence (grey) and presence (blue) of DENV2 NS5. **c** Species level node dated molecular phylogeny with alignments of the STAT2 sequence at residues defined as resistance and susceptibility determinants. Species with STAT2 susceptible to DENV NS5 antagonism are marked (*). Arrows (1) and (2) indicate the acquisition of resistance to the reversion to susceptibility, respectively. **d** Relative IFN signaling (Mean ± s.d, n = 6) in the absence (grey) and presence (blue) of DENV2 NS5 of (i) *P. coquereli* STAT2 with *L. catta* amino acids (ii) *L. catta* STAT2 with *P. coquereli* amino acids (iii) *L. catta* STAT2 with *E. flavifrons* amino acids (iv) *E. flavifrons* STAT2 with *L. catta* amino acids. **e** Relative STAT2 abundance (Mean ± s.d, n = 3) following DENV (blue) or ZIKV (pink) infection compared to the uninfected condition for each cell line. Values are normalized to the percent infection for each virus in each cell line (quantification of Supplementary Fig. 6b, gating strategy and quantification in Supplementary Fig. 8). Data is generated from the quantification of three independent experiments. **f** Relative IFN signaling (Mean ± s.d, n = 6) of STAT2 chimeras with increasing amounts of DENV2 NS5. Values are normalized to signaling in absence of DENV2 NS5 and the baseline is set to reporter levels in the absence of STAT2. Curves are fit using least-squares fit method of non-linear regression to calculate IC50 values for each STAT2 in units of ng of DENV2 NS5 plasmid. Source data are provided as a Source Data file. All IFN signaling values are derived from at least two independent experiments each with three technical replicates.

The evolution of resistance to flavivirus-like antagonism is complex, and in all cases we studied, involves multiple amino acid residues. In each resistant STAT2 sequence, we identified multiple amino acid residues required to disrupt NS5 binding. We believe this is a consequence of flavivirus NS5 proteins (or other anti-immune proteins that function similarly) binding to multiple STAT2 surfaces, some of which overlap with a critical IRF9 interaction site. Although single amino acid changes can disrupt NS5 binding, these also affect IRF9 binding[11] and thus likely are deleterious. Furthermore, it appears that resistance determinants must be acquired in a specific order that maintains STAT2 stability and signaling function, which is consistent with previous work indicating that coevolution of some STAT2 residues likely resulted from impacts to STAT2 functionality[9]. The need for multiple changes in a specific order illustrates the concept of an evolutionary ridge where there is narrow path between two phenotypes (susceptibility and resistance) that must be followed to maintain functionality[16]. While our evolutionary analyses identified rapidly evolving residues that impact susceptibility to NS5 antagonism, there were additional rapidly evolving residues that did not impact this phenotype and suggest that multiple selective pressures are acting on STAT2 evolution.

Our analysis of lemur STAT2 evolution also highlighted that the acquisition of resistance to flavivirus-like antagonism is not unidirectional, as we found that some modern lemurs lost their resistance to DENV NS5 late in their evolution. Even here, we saw that the loss of resistance needed to follow specific evolutionary trajectories to avoid breaking STAT2. This loss of resistance suggests that there may be a cost to being resistant to such antagonism, perhaps in the form altered STAT2 functionality or increased susceptibility to another pathogen that targets STAT2 through other determinants.

We have unraveled evolutionary changes in mammalian STAT2 sequences that impacted STAT2 ability to be antagonized by three primate flaviviruses. However, the precise viruses that exerted the evolutionary pressure to select for these changes are unknown. Since the STAT2 changes are located to the STAT2 domain targeted by three diverse flaviviruses (DENV, ZIKV and SPOV), it is likely that these changes were selected by related flaviviruses circulating in nature when that these changes were acquired. In any case, our results highlight that many non-primate species are susceptible to STAT2 antagonism by flavivirus NS5 proteins and identify them as potential reservoir hosts of these viruses. While further studies are required to definitively demonstrate the capacity of each species to serve as a reservoir host, STAT2 resistance to NS5 antagonism is a significant species tropism determinant for these flaviviruses. While many factors influence the capacity of a species to serve as a reservoir host, knowing the STAT2 susceptibility serves as a first level assessment of susceptibility. Identifying these species is especially important given the tremendous global burden of mosquito-borne flaviviruses[17] and the risk of increased geographical range of these viruses due to rising global temperatures[10].

## Methods

### Cell lines and culture

STAT2 KO HEK-293T cells were grown in Dulbecco's modified Eagle medium (DMEM) (Gibco, 11995065) supplemented with 10% fetal bovine serum (FBS) (Gibco, A5256701) at 37 °C. Huh-7.5 cells[18] (provided by Charles Rice, Rockefeller University, New York, NY) were grown at 37 °C in Dulbecco's modified Eagle medium (DMEM) (Gibco, 11995065) supplemented with 10% fetal bovine serum (FBS) (Gibco, A5256701).

To generate STAT2 KO HEK-293T cells, approximately 400,000 HEK-293T (ATCC, CRL-3216) cells were seeded into 6-well plates in complete DMEM. The next day, cells were transfected with 1 μg of LentiCRISPRv2 plasmid containing a STAT2 targeting sequence (either guide 3 or guide 4) from the Brunello CRISPR library[19]. Two days post transfection, 2 μg/mL puromycin was added to the media. After two days of selection, the media was changed to complete DMEM without puromycin, and surviving cells were allowed to expand. Five days after the completion of selection, single cell clones were obtained by serial dilution and plating in 96-well plates using 50% complete DMEM and 50% HEK-293T cell conditioned DMEM (supernatant from naïve HEK-293T, 0.22 μm filtered), with 20% FBS. Approximately 14 days post seeding, clonal populations were expanded and validated for STAT2 KO. A single validated clone was used in this study.

Primary dermal fibroblasts were prepared from skin samples obtained from the Duke Lemur Center. Using aseptic technique, skin samples were finely minced with a razor blade and incubated for 1 h at 37 °C in a six well plate in 2 mL of Liberase solution (1X DMEM (Gibco, 11995065) supplemented with 0.2% sodium bicarbonate (Gibco, 25080094) and 0.28 Wu/mL Liberase (Roche, 5401119001)). Samples were then diluted with 4 mL of pLF media (DMEM (Gibco, 11995065) supplemented with 10% FBS (Gibco, A5256701), 50 U/mL Pen Strep (Gibco, 15070063), 30 μg/mL Gentamicin and 15 ng/mL Amphotericin (Lonza, CC-4083) and 55 μm beta-mercaptoethanol (Gibco, 21985023)) and incubated overnight at 37 °C. To produce a single cell suspension, 500 μL of EDTA solution (1X PBS (Gibco, 10010023) supplemented with 2.5 mM EDTA (Invitrogen, 15575020) and 2% FBS (Gibco, A5256701) was added to each sample and incubated for five minutes at 37 °C. The samples were then triturated using a pipette to disperse any clumps and then passed through a 70 μm cell strainer ensuring all cells were collected by washing the strainer with PBS. Cells were brought to a total volume of 50 mL in PBS and pelleted by centrifugation at $800 \times g$ for five minutes before plating in a six well plate in 3 mL of pLF. Cells were expanded and frozen stocks were prepared for future use. Cells were grown in DMEM (Gibco, 11995065) supplemented with 10% FBS (Gibco, A5256701), 50 U/mL Pen Strep (Gibco, 15070063), 30 μg/mL Gentamicin and 15 ng/mL Amphotericin (Lonza, CC-4083) and 55 μm beta-mercaptoethanol (Gibco, 21985023).

## Plasmids

All STAT2 and flavivirus NS5 expression vectors were generated by cloning into a modified version of pSCRPSY (GenBank accession no. KT368137.1) (a gift from Paul Bieniasz, Rockefeller University, New York City, NY), termed pSCRPSYneo, which encodes a neomycin resistance cassette in place of PAC2ATagRFP sequence. Each of the STAT2 and NS5 sequences evaluated in this assay were cloned into the XbaI/XhoI restriction sites in the multiple cloning site using the In-Fusion Cloning Kit (Takara 638947) (plasmid and oligo sequences available upon request). All PCR amplified sequences and cloning sites were verified by Sanger sequencing. DNA templates for DENV NS5 proteins were provided by Priya Shaw (University of California Davis). STAT2 sequences were cloned with a C-terminal GFP tag. For co-immunoprecipitation assays, a 3x-FLAG tag (amino acid sequence DYKDHDGDYKDHDIDYKDDDDK) was introduced to the C-terminus of NS5. A list of STAT2 and NS5 sequences can be found in (Supplementary Data 4).

## STAT2 antagonism assay

50,000 STAT2 KO HEK-293T cells were transfected ISG54-ISRE-Fluc (200 ng), TK-Rluc (40 ng), STAT2 (3.7 ng), and NS5/empty vector (varies by antagonist). Antagonist plasmid quantities are as follows; ZIKV MR766 NS5 (33 ng), ZIKV PRVABC59 NS5 (33 ng), DENV1 Western Pacific 74 NS5 (100 ng), DENV2 16681 NS5 (100 ng), DENV3 IN/BID-V2417/1984 NS5 (100 ng), DENV4 341750 NS5 (100 ng), and SPOV SA Ar94 NS5 (100 ng). At 24 h post-transfection, the cells with treated with 100 U of human IFNb (at 100 U/ml). Cells were lysed 24 h after IFNb treatment, and FLuc and RLuc expression was measured using the Dual-Luciferase Reporter Assay System (Promega, E1910) and a BioTek Synergy H1 Multimode Reader (Agilent). Data were calculated as a ratio of Fluc:RLuc to normalize for transfection efficiency. At least two independent experiments with three biological replicates were performed for all reported values. Results of two-sided t-test (p-value) can be found in source data. Statistical significance was determined without correction for multiple comparisons, with alpha = 0.05. Each row was analyzed individually, without assuming a consistent SD.

## Co-immunoprecipitation

500,000 STAT2 KO HEK-293T cells were transfected with 1 µg of either an empty vector or a vector coding for GFP, STAT2-GFP, or NS5-FLAG. 24 h post transfection, cells were lysed in 300 µL of 1X RIPA buffer (Millipore, 20-188) supplemented with cOmplete, Mini Protease Inhibitor Cocktail (Roche, 11836153001). Lysates were clarified by collecting the supernatant following centrifugation at 15,000 g for 15 min. STAT2-NS5 mixtures were then made using equal volumes of the different lysates and a 20 µL aliquot of each mixture was taken and mixed with 20 µL of 2X SDS-PAGE sample buffer plus beta-mercaptoethanol for analysis of the whole cell lysate (WCL). The remaining volume was mixed with 50 µL of EZview Red ANTI-FLAG M2 Affinity Gel (Millipore, F2426) and incubated overnight on an orbital shaker at 4 °C to immunoprecipitate the NS5-FLAG. The affinity gel was then washed six times by addition of 1 mL of 1X RIPA buffer with protease inhibitor, shaking at 4 °C for 10 min, and centrifugation at 10,000 g for 1 min. To elute proteins, 50 µL of 2X SDS-PAGE sample buffer plus beta-mercaptoethanol was added to the affinity gel and samples were heated to 95 °C for 5 min. Equal volumes of each WCL and IP sample were run on 4–20% Mini-PROTEAN® TGX™ Precast Protein Gels (Bio-Rad, #4561096). Proteins were transferred using Trans-Blot Turbo Mini 0.2 µm PVDF Transfer Packs (Bio-Rad, #1704156) immunoblotted with an anti-β-actin antibody directly conjugated horseradish peroxidase (HRP) (Abcam, ab49900) to ensure analysis of equal protein concentrations. Monoclonal ANTI-FLAG M2-HRP (Sigma-Aldrich, A8592) was used to detect NS5-FLAG, and rabbit anti-GFP antibody (Abcam, ab290) was used to detect GFP and GFP STAT2. HRP-conjugated anti-

rabbit secondary antibody (Amersham, NA934V) was used, and detection was performed with SuperSignal West Femto Maximum Sensitivity Substrate (Thermo Scientific, 34095).

## STAT2 immunoblotting

STAT2 KO HEK-293T were transfected with equal amounts of each STAT2 expression or empty vector. 24 h post transfection cells were lysed in 100 µL of 1X SDS-PAGE sample buffer plus beta-mercaptoethanol. Cell lysates were heated to 95 °C for 5 min, and equal volumes of each were run on 4–20% Mini-PROTEAN® TGX™ Precast Protein Gels (Bio-Rad, #4561096). Proteins were transferred using Trans-Blot Turbo Mini 0.2 µm PVDF Transfer Packs (Bio-Rad, #1704156) immunoblotted with an anti-β-actin antibody directly conjugated HRP (Abcam, ab49900) to ensure analysis of equal protein concentrations. A rabbit anti-STAT2 antibody (Invitrogen, #44-362 G) that recognizes human STAT2 c-terminus, which is universal among all tested STAT2 chimeras, was used to probe for STAT2. HRP-conjugated anti-rabbit secondary antibody (Amersham, NA934V) was used, and detection was performed with SuperSignal West Femto Maximum Sensitivity Substrate (Thermo Scientific, 34095).

## Lemur STAT2 RT-PCR and sequencing

To sequence the STAT2 ND and CCD of the lemur species evaluated in this study, we first extracted RNA from blood and/or tissue samples provided by the Duke Lemur Center in Durham, North Carolina using a Total RNA extraction kit (Omega BIO-TEK, R6834-01). RNA extracted from cells was then used for cDNA synthesis using a First Strand cDNA Synthesis Kit (Roche, 11483188001). Using primers which bind to the conserved sequences in the 5′ UTR and coding region of STAT2 mRNA (STAT2 5′ UTR forward primer 5′-GAGACCCTAACCAGAGCCCAA, 5′STAT2 (nt 410) reverse primer 5′-AGCCCTCTGAGCCTGAATCAGAATTC, STAT2 (nt 390) forward primer 5′-TGGCTGAGATGATGTTTAATC, and STAT2 (nt 1025) reverse primer 5′-TCGATGGGGAGTTTGAGGCAT), the STAT2 ND and CCD were amplified from cDNA in two overlapping PCR products (5′UTR-nt410 and nt390-nt1025). These PCR products were then cloned into the pGEM-T-Easy vector (Promega, A1360) and Sanger sequenced.

## Virus titration

The absolute titers of ZIKV and DENV were determined for each cell line used by immunostaining with the pan-flavivirus E protein-reactive 4G2 antibody (Absolute antibody, Ab00230-10.0) and flow cytometry[20]. One day prior to infection, 24-well plates were seeded at a density of 50,000 cells/well. The next day, 1 mL of serially diluted virus in DMEM with 2% FBS was added to each well. Staining was performed one day post infection to avoid viral spread. Infectious titer was calculated using only the viral dilutions leading to 10% to 1% infected cells. Infectious units per ml were calculated using the following formula: percentage of infectious events × number of cells seeded / volume (mL) of viral inoculum. Data were collected on a Gallios flow cytometer (Beckman Coulter) and data were analyzed using FlowJo (v10) and Kaluza Analysis software.

## Endogenous STAT2 degradation experiment

To measure the degradation of endogenous STAT2 following flavivirus infection, 50,000 cells were seeded per well in a 24 well plate, and infected with either ZIKV or DENV at an MOI of 3 infectious units per cell to attempt to infect most of the cells. Despite this calculation, we found it difficult to infect all cells in each condition. Thus, in each experiment an additional well of each cell line was infected with each virus and collected at 24 h post infection and analyzed by flow cytometry to calculate that actual proportion of infected cells. This ratio was used to correct the observed STAT2 degradation levels, to ensure that the STAT2 from uninfected bystander cells did not confound our measurements. Cells were lysed in 75 µL of 1X SDS-PAGE sample buffer

plus beta-mercaptoethanol at 18 h post infection and were analyzed by immunoblot. Blots were quantified using ImageJ (1.54d) to determine changes in STAT2 abundance relative to the uninfected condition. Three independent replicates were performed for statistical analysis, with one representative replicate shown. Flow data were collected on a Gallios flow cytometer (Beckman Coulter) and data were analyzed using FlowJo (v10) and Kaluza Analysis software.

## Generation species phylogeny
To infer species level relatedness among the mammals investigated in this study, we generated phylogenetic trees using Mesquite[21] subsets of data from a previously published node-dated, maximum clade credibility tree[22].

## Rapid evolution analysis
STAT2 sequences were obtained from GenBank (rodents, prosimians), RT-PCR of tissue samples (prosimians), and from the Zoonomia project (bats)[23]. Multiple sequence alignments were generated using MUSCLE as implemented in MEGA X[24]. Gaps in alignments were trimmed using Gblocks[25] and alignments were further curated by manual inspection. Species trees were inferred using TimeTree (rodents, prosimians)[26] or obtained from Zoonomia (bats). PAML was used to assess signatures of evolutionary pressure present in STAT2 ND/CCD domains. Briefly, CodeML was used with the F3x4 codon frequency table and default settings[13]. Likelihood ratio tests were used to compare Model 8 (beta and omega—allowing for positive selection) and Model 7 (beta—no positive selection), as well as Model 8 and Model 8a (beta and omega, constrained to no positive selection). Sites that passed a stringent test (Bayes Empirical Bayes) were annotated as being under positive selection. To complement PAML analyses, FUBAR[12] was used as implemented on DataMonkey[27], with species trees as references. To determine statistical significance, a posterior probability of 0.95 for PAML analyses and 0.9 for FUBAR. To detect evidence of episodic diversification, a free-ratio analysis (branch-site model) was performed in PAML with Model 1 (branch) and NSsites = 0 to obtain a dN/dS value for each branch. A complementary analysis was performed using ABSREL[28] as implemented in DataMonkey, using default settings.

## Generation of LentiCRISPRv2 STAT2 plasmids
STAT2-targeting sequences from the Brunello Human CRISPR Knockout Pooled Library (Addgene #73179) were cloned into the LentiCRISPRv2 plasmid (Addgene #52961) using protocols from the Zhang lab[29]. Briefly, LentiCRISPRv2 was digested with BsmBI-v2 (NEB) and treated with rAPid Alkaline Phosphatase (Roche). Oligos containing STAT2-targeting sequences and plasmid overhangs were annealed and phosphorylated with T4 PNK (NEB) and ligated into the digested plasmid with T4 DNA ligase (NEB).

Oligo sequences are written 5′ to 3′. STAT2-targeting sequences are in uppercase, plasmid overhangs are in lowercase. guide 3 forward primer (caccgAGCAACATGAGATTGAATCC), guide 3 reverse primer (aaacGGATTCAATCTCATGTTGCTc), guide 4 forward primer (aaacACCCAGTTGGCTGAGATGATc), guide 4 reverse primer (caccgATCATCTCAGCCAACTGGGT).

## Reporting summary
Further information on research design is available in the Nature Portfolio Reporting Summary linked to this article.

## Data availability
The STAT2 sequence data generated in this study have been deposited on NCBI Genbank (*Cheirogaleus medius* (PP600020), *Varecia variegata* (PP600021), *Varecia rubra* (PP600022), *Propithecus tattersalli* (PP600023), *Mirza zaza* (PP600024), *Eulemur rufus* (PP600025), *Eulemur mongoz* (PP600026), *Eulemur flavifrons* (PP600027), *Daubentonia madagascariensis* (PP600028)). Source data including unprocessed western blot images are provided with this paper. All biological materials including plasmids and cells lines will be made available for research purposes upon request. Source data are provided with this paper.

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

## Acknowledgements

This study was supported in part by NIH grants R01 AI175303 (M.J.E. and A.G.S.), and R35 GM134936 (N.C.E.). E.C.V. and R.B.R. were supported by T32 (AI007647). C.K. was supported by T32 (AI083203). The Duke Lemur Center (Durhan, NC) kindly provided lemur blood and skin samples (Duke Lemur Center publication #1589). The authors want to thank Priya Shah (University of California Davis) for kindly provided DENV NS5 expression vectors; Charles Rice (Rockefeller University, New York, NY) for kindly providing Huh-7.5 cells; Paul Bieniasz (Rockefeller University, New York City, NY) for kindly providing the pSCRPSY vector; Ryan Langlois (University of Minnesota) for protocols and technical advice on fibroblast isolation; and Jesse Bloom (Fred Hutchinson Cancer Center, Seattle, WA) and Lisa Miorin (Icahn School of Medicine at Mount Sinai, New York City, NY) for helpful discussions in the development of this project. The funders had no role in study design, data collection and analysis, decision to publish, or preparation of the manuscript.

## Author contributions

E.C.V. and M.J.E. conceived this study. E.C.V., M.S.S., M.J.J., R.B.R., I.N.B., M.Q., E.A.B., E.B., R.E.H., and C.K. collected, analyzed, and interpreted data. M.J.E. and N.C.E. supervised this study. M.J.E. and A.G.S. acquired funding for the project. E.C.V. and M.J.E. wrote the manuscript. All authors reviewed the results and approved the final version of the manuscript.

## Competing interests

Adolfo García-Sastre has received research support from GSK, Pfizer, Senhwa Biosciences, Kenall Manufacturing, Blade Therapeutics, Avimex, Johnson & Johnson, Dynavax, 7Hills Pharma, Pharmamar, ImmunityBio, Accurius, Nanocomposix, Hexamer, N-fold LLC, Model Medicines, Atea Pharma, Applied Biological Laboratories and Merck, outside of the reported work. A.G.-S. has consulting agreements for the following companies involving cash and/or stock: Castlevax, Amovir, Vivaldi Biosciences, Contrafect, 7Hills Pharma, Avimex, Pagoda, Accurius, Esperovax, Farmak, Applied Biological Laboratories, Pharmamar, CureLab Oncology, CureLab Veterinary, Synairgen, Paratus, Pfizer and Prosetta, outside of the reported work. A.G.-S. has been an invited speaker in meeting events organized by Seqirus, Janssen, Abbott and Astrazeneca. A.G.-S. is inventor on patents and patent applications on the use of antivirals and vaccines for the treatment and prevention of virus infections and cancer, owned by the Icahn School of Medicine at Mount Sinai, New York, outside of the reported work. The remaining authors declare no competing interests.
