## [Peer Review File · Nature Communications]

Evolution of STAT2 resistance to flavivirus NS5 occurred multiple times despite genetic constraintsREVIEWER COMMENTS

Reviewer #1 (Remarks to the Author):

The authors presented exemplary research evaluating the effectiveness of amino acid combinations within the STAT2 molecule in several mammalian species. This investigation is particularly notable due to the observed variations in susceptibility to infection by prevalent flaviviruses and the correlated clinical outcomes among different mammalian taxa. However, a few suggestions and corrections can improve readers' understanding of the results.

1-The authors inform in lines 62-65: "Previous studies reported that STAT2 in mammals has evolved under positive selection⁹ suggesting that it may be engaged in one or more host-pathogen genetic conflicts, but the impact that the changes in STAT2 sequence acquired during the evolution of mammals on NS5 antagonism has not been investigated."

Comment: The authors reference Landau et al.'s work (2022)⁹ and subsequently assert that "...the impact that the changes in STAT2 sequence acquired during the evolution of mammals on NS5 antagonism has not been investigated." This assertion is inaccurate, as the cited paper extensively examines this topic. For instance, Landau et al. (2022) showed that rodents currently have an edge in combating infections caused by well-known flaviviruses (ZIKA, DENV, and YFV), especially when compared to primates, and that the simultaneous maintenance of adequate cellular function while subjected to selective pressure to combat systematic viral attacks produces distinct evolutionary patterns in STAT2. Furthermore, Landau et al. (2022) suggested that the STAT2 patterns, which include epistatic combinations of amino acids, in these mammals were probably motivated by clashes with other flaviviruses, but not necessarily those that are currently prevalent in human infections. Their results point to dynamics that fit with a molecular evolutionary scenario shaped by a thought-provoking virus-host arms race.

What remains unaddressed is the absence of functional studies. Additionally, the manuscript by Evans et al. also presents results about the diversity pattern of STAT2 in bats and lemurs, which have not been previously studied or evaluated. Therefore, the authors should duly acknowledge the existing literature and focus on advancing their study by incorporating functional analyses, among other findings, thereby contributing to a broader understanding of this field, without needing to ignore previous studies.

2- The absence of scientific species names within the main text is deemed inappropriate in a scientific article. Even within supplementary materials, where datasets are more extensively presented, the omission of specific species' scientific names persists. For instance, the authors reference "marmoset" without delineating the particular species. Given the considerable diversity among marmoset species, such ambiguity compromises the requisite precision and clarity indispensable for scientific discourse. Rodents could be referred to in the table as the order Rodentia and "bats" as species of the order Chiroptera; in short, more precision in the phylogenetic classification is essential for a scientific article of this nature. The authors should fix this flaw.

3- The authors utilize two methods to present the results of STAT2 expression's ability to mediate signaling via IFN: one in Figure 1f-h and the other in Expanded Data Table 1. In the former, relative IFN signaling is illustrated, defined as the activity rate of the two luciferases normalized by naïve HEK-293T cells transfected with reporters only, alongside STAT2 fold-change, calculated by dividing the mean of relative IFN signaling in the absence of the antagonist by the mean of relative IFN signaling in the presence of the viral protein NS5. This dual approach to presenting results may be confusing to the reader. Standardization or clearer delineation in the main text could enhance comprehension, especially since figures and expanded data tables are cited in the same context.

4- The authors inform in lines 102-103: “The only notable difference between 103 ZIKV and SPOV NS5 in this assay was that beaver STAT2 was more susceptible to SPOV.”

Comment: There are other situations worthy of note, for example the case of the white faced capuchin (*Cebus capucinus*?) but in the opposite direction. What is the criteria for getting the beaver's attention? It seems to be the nature of difference rather than just the magnitude of the difference.

Castor canadensis or Castor fiber?

5- As for evolutionary analyses, using FUBAR, the authors also have access to sites under severe constriction due to powerful purifying selection; this could be. Positive selection signal may be masked if PALM analyzes only consider species of clades with phylogenetic solid signal (e.g., order Rodentia, “Bats”); thus, they can share functional variants. Therefore, species from other clades should be considered. The PALM package also has a CODEML branch site capable of detecting positive selection along a specific tree branch.

Reviewer #2 (Remarks to the Author):

Synopsis: This study provides an analysis of dengue 2, Zika, and Spondweni virus NS5-mediated inhibition of STAT2 signaling using an extensive array of mammalian STAT2 sequences. Initial studies were conducted in HEK293T expressing STAT2 from multiple primate, rodent, bat, and other mammals. Notably, most rodent species examined were resistant to NS5-mediated STAT2 degradation. Resistance was mapped to a 12 amino acid region in the STAT2 coiled-coil domain (aa 198-209, differing by 8 amino acids between human and mouse), which also mediates binding of NS5 and STAT2. In contrast, guinea pig STAT2 is also resistant to NS5 degradation, but had a CCD sequence that predicts susceptibility. Resistance in the guinea pig was mapped to the N-terminal domain of STAT2, indicating a 2nd, independent, evolutionary pathway to NS5 resistance in this rodent. A similar analysis was also performed using STAT2 derived from several bat species. All species were susceptible to ZIKV NS5, but showed variable sensitivity to DENV NS5, allowing swapping of domains between STAT2 proteins and mapping of resistance determinants. The third

analysis focused on species of lemurs. The authors initially observed that *P. coquereli* was resistant to DENV NS5 inhibition of STAT2 signaling, which is unusual in primate species. Further analysis of lemur species demonstrated that some species are susceptible and others resistant. Comparison of sequences associated with resistance with the lemur phylogenetic tree suggests that resistance was acquired ~60 million years ago and later lost in some species. Infection of dermal fibroblasts derived from several resistant and susceptible lemur species was largely consistent with the results of exogenously expressed STAT2 and NS5, although *P. coquereli* did show some STAT2 degradation. Throughout this study, the authors apply 2 previously described methods of inferring if specific residues are under evolutionary positive selection. Although the rodent analysis did not show any STAT2 residues associated with NS5 resistance under positive selection, several within bat and lemur species did show this association.

Comments and concerns:

1. This study provides an extensive analysis of mammalian STAT2 susceptibility to ZIKV, DENV, and SPO NS5-mediated degradation and inhibition of signaling. Experiments identifying resistant/susceptible STAT2s and mapping of resistance determinants are well executed and the data is convincing.
2. In figure 5e and ext. data 6b, the authors demonstrate STAT2 degradation in DENV or ZIKV infected fibroblasts derived from different lemur species. Do the viruses grow with similar kinetics in these cells, or is DENV replication restricted in cells with resistant STAT2? Are these cells equally susceptible to infection? The methods section notes that percentage of infected cells was determined by flow cytometry, but this data is not presented in the manuscript.
3. The overarching goal of this manuscript is to test the hypothesis that STAT2 evolution is driven by resistance to ZIKV/ DENV (or an earlier proto-flavivirus) NS5-mediated binding and degradation. The data from the bat and lemur analysis suggest such selective pressure is possible, but the FUBAR/ PAML analyses identify other residues not associated with NS5 as also under positive selective pressure, suggesting that other factors are also driving STAT2 evolution.
4. Although expressing STAT2 in 293 cells deleted of endogenous STAT2 provides an optimal system for evaluating the STAT2/ NS5 interaction, further data should be provided about the relative susceptibility of individual species (or cells derived from those species) to ZIKV/ DENV/ SPOV infection. Although not feasible for all species tested, this could be provided for the lemur fibroblasts described in Fig 5.
5. The suggestion that STAT2 susceptibility may indicate a potential reservoir species (line 325) is not supported. For example, even flaviviruses that are capable of inhibiting interferon signaling in rodents (e.g. WNV) are restricted by other host factors, such as OAS2b. Because multiple virus-host interactions influence suitability of a species to be a reservoir, the impact of these studies, while intriguing, is limited.

Response to Reviewers

We thank the reviewers for the constructive comments that helped us strengthen this resubmission. We offer the following specific responses to reviewer questions. In this letter, the original reviewers' comments are shown in bold text and our responses are in regular text.

Reviewer #1:

The authors presented exemplary research evaluating the effectiveness of amino acid combinations within the STAT2 molecule in several mammalian species. This investigation is particularly notable due to the observed variations in susceptibility to infection by prevalent flaviviruses and the correlated clinical outcomes among different mammalian taxa. However, a few suggestions and corrections can improve readers' understanding of the results.

1-The authors inform in lines 62-65: "Previous studies reported that STAT2 in mammals has evolved under positive selection⁹ suggesting that it may be engaged in one or more host-pathogen genetic conflicts, but the impact that the changes in STAT2 sequence acquired during the evolution of mammals on NS5 antagonism has not been investigated."

Comment #1: The authors reference Landau et al.'s work (2022) and subsequently assert that "...the impact that the changes in STAT2 sequence acquired during the evolution of mammals on NS5 antagonism has not been investigated." This assertion is inaccurate, as the cited paper extensively examines this topic. For instance, Landau et al. (2022) showed that rodents currently have an edge in combating infections caused by well-known flaviviruses (ZIKA, DENV, and YFV), especially when compared to primates, and that the simultaneous maintenance of adequate cellular function while subjected to selective pressure to combat systematic viral attacks produces distinct evolutionary patterns in STAT2. Furthermore, Landau et al. (2022) suggested that the STAT2 patterns, which include epistatic combinations of amino acids, in these mammals were probably motivated by clashes with other flaviviruses, but not necessarily those that are currently prevalent in human infections. Their results point to dynamics that fit with a molecular evolutionary scenario shaped by a thought-provoking virus-host arms race.

What remains unaddressed is the absence of functional studies. Additionally, the manuscript by Evans et al. also presents results about the diversity pattern of STAT2 in bats and lemurs, which have not been previously studied or evaluated. Therefore, the authors should duly acknowledge the existing literature and focus on advancing their study by incorporating functional analyses, among other findings, thereby contributing to a broader understanding of this field, without needing to ignore previous studies.

We apologize for not fully acknowledging the prior work done in Landau et al. (2022). We have modified our manuscript on lines 62-67 & 319-321 (lines 62-68 & 336-340 of track changes document) to correct this oversight.

2- The absence of scientific species names within the main text is deemed inappropriate in a scientific article. Even within supplementary materials, where datasets are more extensively presented, the omission of specific species' scientific names persists. For instance, the authors reference "marmoset" without delineating the particular species. Given the considerable diversity among marmoset species, such ambiguity compromises the requisite precision and clarity indispensable for scientific discourse. Rodents could be referred to in the table as the order Rodentia and "bats" as species of the order

Chiroptera; in short, more precision in the phylogenetic classification is essential for a scientific article of this nature. The authors should fix this flaw.

We completely agree that precise use of scientific species names is important. We have addressed this reviewer comment by modifying the relevant figures and text to utilize only scientific names for the species involved.

3- The authors utilize two methods to present the results of STAT2 expression's ability to mediate signaling via IFN: one in Figure 1f-h and the other in Expanded Data Table 1. In the former, relative IFN signaling is illustrated, defined as the activity rate of the two luciferases normalized by naive HEK-293T cells transfected with reporters only, alongside STAT2 fold-change, calculated by dividing the mean of relative IFN signaling in the absence of the antagonist by the mean of relative IFN signaling in the presence of the viral protein NS5. This dual approach to presenting results may be confusing to the reader. Standardization or clearer delineation in the main text could enhance comprehension, especially since figures and expanded data tables are cited in the same context.

We appreciate this reviewer's suggestion and apologize for the confusion. To be consistent throughout the manuscript as well as presenting the data in as straight forward a manner as possible, we have removed Extended Data Table 1 from the manuscript.

4- The authors inform in lines 102-103: "The only notable difference between ZIKV and SPOV NS5 in this assay was that beaver STAT2 was more susceptible to SPOV".

Comment: There are other situations worthy of note, for example the case of the white faced capuchin (*Cebus capucinus*?) but in the opposite direction. What is the criteria for getting the beaver's attention? It seems to be the nature of difference rather than just the magnitude of the difference. *Castor canadensis* or *Castor fiber*?

Our original criteria for highlighting differences between ZIKV and SPOV antagonism of *Castor canadensis* STAT2 were based on the degree of signaling observed in the presence of each NS5. *Castor canadensis* STAT2 could still mediate appreciable signaling in the presence of ZIKV NS5, but signaling was reduced 27-fold in the presence of SPOV NS5 compared to when no antagonist was present. Since this is a rather nuanced way to interpret the data, we have altered the main text at lines 105-106 (lines 106-114 of the tracked changes document) to merely state that the ZIKV and SPOV NS5 patterns of STAT2 antagonism were very similar.

5- As for evolutionary analyses, using FUBAR, the authors also have access to sites under severe constriction due to powerful purifying selection; this could be. Positive selection signal may be masked if PALM analyzes only consider species of clades with phylogenetic solid signal (e.g., order Rodentia, "Bats"); thus, they can share functional variants. Therefore, species from other clades should be considered. The PALM package also has a CODEML branch site capable of detecting positive selection along a specific tree branch.

To improve our study, we have added additional material, contained in Supplementary Data 1-3, that provide the raw data for the FUBAR analyses of the three different mammalian lineages. These data do show evidence of purifying selection at various STAT2 residues but given the large number of residues evolving under purifying selection, we did not address these directly in the main text of the manuscript. We have also added Supplementary Fig. 3, that contains the results of PAML branch site analysis for the three mammalian lineages analyzed in this study. Regarding the selection of species for PAML analysis, we chose a tree length with a defined clade to avoid

the possibility of saturation of synonymous mutations that could give misleading results. Doing analysis on several clades separately and looking for overlapping sites is of course possible, but we believe this is beyond the scope of the analysis considered necessary for this study.

Reviewer #2

Synopsis: This study provides an analysis of dengue 2, Zika, and Spondweni virus NS5-mediated inhibition of STAT2 signaling using an extensive array of mammalian STAT2 sequences. Initial studies were conducted in HEK293T expressing STAT2 from multiple primate, rodent, bat, and other mammals. Notably, most rodent species examined were resistant to NS5-mediated STAT2 degradation. Resistance was mapped to a 12 amino acid region in the STAT2 coiled-coil domain (aa 198-209, differing by 8 amino acids between human and mouse), which also mediates binding of NS5 and STAT2. In contrast, guinea pig STAT2 is also resistant to NS5 degradation, but had a CCD sequence that predicts susceptibility. Resistance in the guinea pig was mapped to the N-terminal domain of STAT2, indicating a 2nd, independent, evolutionary pathway to NS5 resistance in this rodent. A similar analysis was also performed using STAT2 derived from several bat species. All species were susceptible to ZIKV NS5, but showed variable sensitivity to DENV NS5, allowing swapping of domains between STAT2 proteins and mapping of resistance determinants. The third analysis focused on species of lemurs. The authors initially observed that *P coquereli* was resistant to DENV NS5 inhibition of STAT2 signaling, which is unusual in primate species. Further analysis of lemur species demonstrated that some species are susceptible and others resistant. Comparison of sequences associated with resistance with the lemur phylogenetic tree suggests that resistance was acquired ~60 million years ago and later lost in some species. Infection of dermal fibroblasts derived from several resistant and susceptible lemur species was largely consistent with the results of exogenously expressed STAT2 and NS5, although *P coquereli* did show some STAT2 degradation. Throughout this study, the authors apply 2 previously described methods of inferring if specific residues are under evolutionary positive selection. Although the rodent analysis did not show any STAT2 residues associated with NS5 resistance under positive selection, several within bat and lemur species did show this association.

Comments and concerns:

1. This study provides an extensive analysis of mammalian STAT2 susceptibility to ZIKV, DENV, and SPO NS5-mediated degradation and inhibition of signaling. Experiments identifying resistant/ susceptible STAT2s and mapping of resistance determinants are well executed and the data is convincing.

We appreciate this compliment of our efforts.

2. In figure 5e and ext. data 6b, the authors demonstrate STAT2 degradation in DENV or ZIKV infected fibroblasts derived from different lemur species. Do the viruses grow with similar kinetics in these cells, or is DENV replication restricted in cells with resistant STAT2? Are these cells equally susceptible to infection? The methods section notes that percentage of infected cells was determined by flow cytometry, but this data is not presented in the manuscript.

Per this reviewer's request, we have added Supplementary Fig. 8 that contains the gating strategy and results of the flow cytometry quantification infection for the STAT2 degradation experiment. We agree that using the lemur derived fibroblasts to investigate the impact of STAT2 resistance on viral replication would be a powerful addition to our study. Indeed, this was the intention when we derived these cells. However, technical issues with these cells prevented their use in such experiments. To our surprise, we found that many of the lemur cells were not responsive to exogenous treatment with either human or universal interferon, which limits our ability to modulate the interferon response equally across cells from different species and assess its impact on viral infection. Additionally, while we showed in our STAT2 degradation experiment that these cells were susceptible to infection, many were not able to support infectious virus production in a manner that appeared independent of STAT2 susceptibility. Thus, these cells were not suitable for multicycle growth curves. This finding actually supports this reviewer's Comment 5, where they state "multiple virus-host interactions influence suitability of a species to be a reservoir". STAT2 is only one of many host factors that impact flavivirus species tropism.

3. The overarching goal of this manuscript is to test the hypothesis that STAT2 evolution is driven by resistance to ZIKV/ DENV (or an earlier proto-flavivirus) NS5-mediated binding and degradation. The data from the bat and lemur analysis suggest such selective pressure is possible, but the FUBAR/ PAML analyses identify other residues not associated with NS5 as also under positive selective pressure, suggesting that other factors are also driving STAT2 evolution.

We agree with this reviewer that STAT2 evolution is impacted by many factors. Our goal was to explore how STAT2 sequence differences between species impacts flavivirus antagonism efficiency as a means to test our hypothesis that flaviviruses are one such factor. We have modified our discussion at lines 323-326 (lines 343-345 of the tracked changes document) to acknowledge that multiple factors must be influencing STAT2 evolution given we identified rapidly evolving residues that do not impact flavivirus NS5 antagonism.

4. Although expressing STAT2 in 293 cells deleted of endogenous STAT2 provides an optimal system for evaluating the STAT2/ NS5 interaction, further data should be provided about the relative susceptibility of individual species (or cells derived from those species) to ZIKV/ DENV/ SPOV infection. Although not feasible for all species tested, this could be provided for the lemur fibroblasts described in Fig 5.

For the reasons outlined in our above response to Comment 2, we have found the lemur fibroblasts difficult to work with and we were unable to generate data other than what is already shown in our manuscript to address the reviewer's question regarding flavivirus infection in these cells.

5. The suggestion that STAT2 susceptibility may indicate a potential reservoir species (line 325) is not supported. For example, even flaviviruses that are capable of inhibiting interferon signaling in rodents (e.g. WNV) are restricted by other host factors, such as OAS2b. Because multiple virus-host interactions influence suitability of a species to be a reservoir, the impact of these studies, while intriguing, is limited.

We fully agree that STAT2 is only one of many host factors that impact flavivirus species tropism. We have modified this prior statement at lines 341-345 (lines 360-364 of the tracked changes document) to state this assertion more clearly.

REVIEWERS' COMMENTS

Reviewer #1 (Remarks to the Author):

I am pleased to report that the authors have thoroughly addressed all the concerns and pending issues I raised in my review. I now consider the manuscript, "Evolutionary pathways to resistance to flavivirus STAT2 antagonism" by Evans et al., 2024, to be suitable for publication in Nature Communications.

I appreciate the opportunity to contribute to the review process for this stimulating work.

Reviewer #2 (Remarks to the Author):

The revisions in this version of the manuscript adequately address my previous concerns.